# Minimum entropy production by microswimmers with internal dissipation

Abdallah Daddi-Moussa-Ider [1], Ramin Golestanian [1,2] & Andrej Vilfan [1,3] ✉

The energy dissipation and entropy production by self-propelled microswimmers differ profoundly from passive particles pulled by external forces. The difference extends both to the shape of the flow around the swimmer, as well as to the internal dissipation of the propulsion mechanism. Here we derive a general theorem that provides an exact lower bound on the total, external and internal, dissipation by a microswimmer. The problems that can be solved include an active surface-propelled droplet, swimmers with an extended propulsive layer and swimmers with an effective internal dissipation. We apply the theorem to determine the swimmer shapes that minimize the total dissipation while keeping the volume constant. Our results show that the entropy production by active microswimmers is subject to different fundamental limits than the entropy production by externally driven particles.

Microswimmers are microscale objects that move in a self-propelled way through a fluid medium at low Reynolds numbers where viscous forces dominate over inertia[1–3]. They comprise living swimmers such as microorganisms and sperm cells, which have been a subject of keen interest by pioneers of twentieth-century fluid physicists such as Ludwig Prandtl[4] and G.I. Taylor[5], as well as artificially manufactured colloidal microswimmers[6–9]. A central question in the field of microswimmers is the energetic cost of their propulsion. The energetic efficiency of a microswimmer is typically characterized by Lighthill's efficiency[10], defined as the equivalent power needed to pull the swimmer through the fluid with an external force, divided by the actually dissipated power of the active swimmer moving with the same velocity. In biological swimmers, the question is whether and how the swimmers have evolved to achieve a high propulsion efficiency and how close they can come to the theoretical limits set by the laws of hydrodynamics. Although Purcell[11] concluded that the energetic expenditure for swimming represents a very small fraction of the total consumption of bacteria, it is now known that larger microorganisms, like *Paramecium* can use about half of their total power for propulsion[12]. The efficiency of artificial swimmers is still lagging far behind their natural counterparts and its improvement is one of the key challenges on the way towards future technological or biomedical applications. Finally, the entropy production in suspensions of microswimmers is a fundamental question in stochastic thermodynamics and statistical physics[13–15]. A common assumption in these

works is to estimate the "housekeeping" work needed to propel active Brownian particles by representing autonomous propulsion as external forces acting on the particles[16–18], which can be coupled to a chemical reaction[19]. An outstanding question is whether there are other fundamental limits on the entropy production by active particles—both because of their propulsion mechanism and because the laws of self-propelled swimming differ from the externally driven particles. At the core of all these diverse research topics, there is a common fundamental question: what is the minimum amount of dissipation needed by a self-propelled microswimmer, how can it be achieved and how does it compare to the dissipation by a passive object that is pulled through the fluid by an external force.

Many biological microswimmers achieve self-propulsion by performing nonreciprocal deformation cycles via periodic beating of cilia or flagella, slender appendages anchored to their cell body. The waveform assumed by beating flagella or cilia follows an asymmetric pattern in an irreversible fashion to generate propulsion. While bacteria and flagellates usually swim with a small number of long flagella, numerous other biological swimmers such as *Paramecium* or *Volvox* swim by means of thousands of cilia packed on their surfaces. Because the cilia are usually an order of magnitude shorter than the size of the body, they can be described as surface-driven and the action of cilia consists in generating an effective slip velocity along the surface[10,20–24].

For surface-driven microswimmers, the power being dissipated by viscosity can broadly be split, from a hydrodynamic perspective, into

[1]Max Planck Institute for Dynamics and Self-Organization (MPI-DS), 37077 Göttingen, Germany. [2]Rudolf Peierls Centre for Theoretical Physics, University of Oxford, Oxford OX1 3PU, UK. [3]Jožef Stefan Institute, 1000 Ljubljana, Slovenia. ✉e-mail: andrej.vilfan@ds.mpg.de

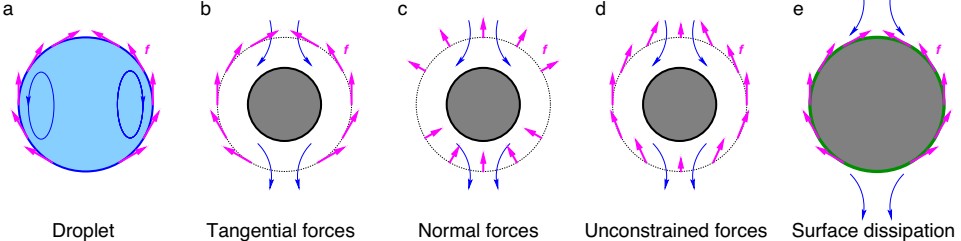

**Fig. 1 | Five scenarios of internal dissipation in an active swimmer. a** An active droplet with a fluid-fluid interface, driven by tangential forces at the interface. **b** A swimmer driven by tangential forces at an outer surface (dotted line). **c** A swimmer driven by normal forces at an outer surface. **d** A swimmer driven by unconstrained forces, allowing both tangential and normal components, at an outer surface. **e** A surface-driven swimmer with internal dissipation in the surface layer (green). The magenta arrows denote the active tractions, defined as the forces exerted by the fluid on the swimmer.

## Table 1 | Minimum dissipation theorems

| Dissipation function | Motivation | Theorem | *V*-problem | *F*-problem |
|---|---|---|---|---|
| External dissipation[38] | Ideal swimmer | $P_A \geq \mathbf{V}_A \cdot \left(\mathbf{R}_{PS}^{-1} - \mathbf{R}_{NS}^{-1}\right)^{-1} \cdot \mathbf{V}_A$ | Perfect slip | No-slip |
| Surface-driven droplet (Fig. 1a) | Active droplets (Marangoni) | $P_A \geq \mathbf{V}_A \cdot \left(\mathbf{R}_{Droplet}^{-1} - \mathbf{R}_{NS}^{-1}\right)^{-1} \cdot \mathbf{V}_A$ | Droplet | No-slip |
| External tangential forces (Fig. 1b) | Model of cilia | $P_A \geq \mathbf{V}_A \cdot \left(\mathbf{R}_{NSi}^{-1} - \mathbf{R}_{CP}^{-1}\right)^{-1} \cdot \mathbf{V}_A$ | No-slip core | No-slip core, no tangential slip shell |
| External normal forces (Fig. 1c) | Phoretic swimmers | $P_A \geq \mathbf{V}_A \cdot \left(\mathbf{R}_{NSi}^{-1} - \mathbf{R}_{DC}^{-1}\right)^{-1} \cdot \mathbf{V}_A$ | No-slip core | No-slip core, zero normal velocity shell |
| External unconstrained forces (Fig. 1d) | Idealized external propulsion | $P_A \geq \mathbf{V}_A \cdot \left(\mathbf{R}_{NSi}^{-1} - \mathbf{R}_{NS}^{-1}\right)^{-1} \cdot \mathbf{V}_A$ | No-slip core | No-slip (outer shell) |
| Surface dissipation (Fig. 1e) | Coarse-grained (cilia, phoretic) | $P_A \geq \mathbf{V}_A \cdot \left(\mathbf{R}_{Navier}^{-1} - \mathbf{R}_{NS}^{-1}\right)^{-1} \cdot \mathbf{V}_A$ | Navier slip | No-slip |

two distinct parts: internal and external. The internal dissipation accounts for the local losses occurring at the propulsive layer. For example, this could be the dissipation within the ciliary layer or within the boundary layer of phoretic swimmers, or the dissipation inside the droplet in the case of self-propelled droplets. The inner dissipation plays the dominant role in most microswimmers, notably in ciliated microorganisms[25–27] and also in phoretic swimmers[28,29]. The external dissipation results from the interaction of the swimmer with the surrounding fluid environment. External dissipation is inevitable as the swimmer displaces the fluid it moves through and is independent of the specifics of the propulsion mechanism. The minimum amount of external dissipation needed by a swimmer of a given shape with a given swimming velocity therefore represents a fundamental problem of low-Reynolds-number hydrodynamics that has been solved analytically for spherical[30–33] and spheroidal swimmers[25,34,35], as well as numerically for arbitrary axisymmetric shapes[36,37]. The propulsive motion can either be stationary[25,34,36] or periodic in time, representing a squirming motion or the motion of the ciliary envelope[30,33,37]. More lately, the swimming efficiency in non-Newtonian fluids has also been investigated[32,35].

We recently derived a general solution to determine the lower bound on external dissipation by an active microswimmer, which we could express with the passive hydrodynamic drag coefficients of two bodies of the same shape: one with the no-slip and one with the perfect-slip boundary condition[38]. The solution also shows that the flow profile of the optimal swimmer is a linear superposition of the flow fields induced by these two passive bodies. The optimal velocity profile, which in principle poses a complex quadratic optimization problem, is then reduced to the solution of two passive flows. By means of this theorem, we determined the flow field of an optimal swimmer of nearly spherical shape using a perturbative analytical approach[39].

In this paper, we show that the approach can be generalized to derive fundamental limits on the total dissipation by a swimmer, comprising both external and internal contributions. We derive novel minimum dissipation theorems for different classes of swimmers, i.e., surface-driven droplets, swimmers with a finite propulsive layer, as well as swimmers with an effective surface dissipation. The formulations of the active problems we study here are illustrated in Fig. 1 and the corresponding theorems are summarized in Table 1. We thus demonstrate that the analytical approach can generically be applied to a broad class of minimum dissipation problems in microswimmers with realistically modeled propulsion mechanisms.

## Results
### Minimum dissipation theorems with internal dissipation
We consider a microswimmer that self-propels through an incompressible viscous fluid. The fluid velocity $\mathbf{v}(\mathbf{x})$ satisfies the Stokes equation $\nabla \cdot \boldsymbol{\sigma} = \mathbf{0}$ together with the continuity equation $\nabla \cdot \mathbf{v} = 0$ where $\boldsymbol{\sigma} = -p\mathbf{I} + 2\mu\mathbf{E}$ is the stress tensor, $p$ the pressure, $\mu$ the shear viscosity, and $\mathbf{E} = \frac{1}{2}\left(\nabla\mathbf{v} + \nabla\mathbf{v}^{\top}\right)$ the rate-of-strain tensor. The swimmer moves with a translational velocity $\mathbf{V}_A$ and angular velocity $\boldsymbol{\Omega}_A$, which can be treated together as a rigid body velocity described by the 6-component vector $\mathbf{V}_A = [\mathbf{V}_A, \boldsymbol{\Omega}_A]$. Likewise the total force $\mathbf{F}_A$ and torque $\mathbf{M}_A$ exerted by the fluid on the swimmer can be merged to a generalized force $\mathbf{F}_A = [\mathbf{F}_A, \mathbf{M}_A]$. In the absence of external (other than hydrodynamic) forces, the swimmer is force- and torque-free, $\mathbf{F}_A = \mathbf{0}$. The swimmer self-propels by imposing a fluid velocity $\tilde{\mathbf{v}}$ on its surface. Here, all velocities in the co-moving frame are denoted as $\tilde{\mathbf{v}}$ and those in the laboratory frame as $\mathbf{v}$. Alternatively, depending on the formulation, the swimmer can also impose an active contribution to the force (traction) density on its surface $\mathbf{f}_A$. As the swimmer moves through the fluid, it dissipates the power $P_A$. The dissipation consists of an external contribution in the fluid outside the swimmer and an internal contribution either in fluids inside the swimmer or internally in the flow-generating mechanism.

In the following, we summarize the minimum dissipation theorems that will be derived in the subsequent sections. All theorems take

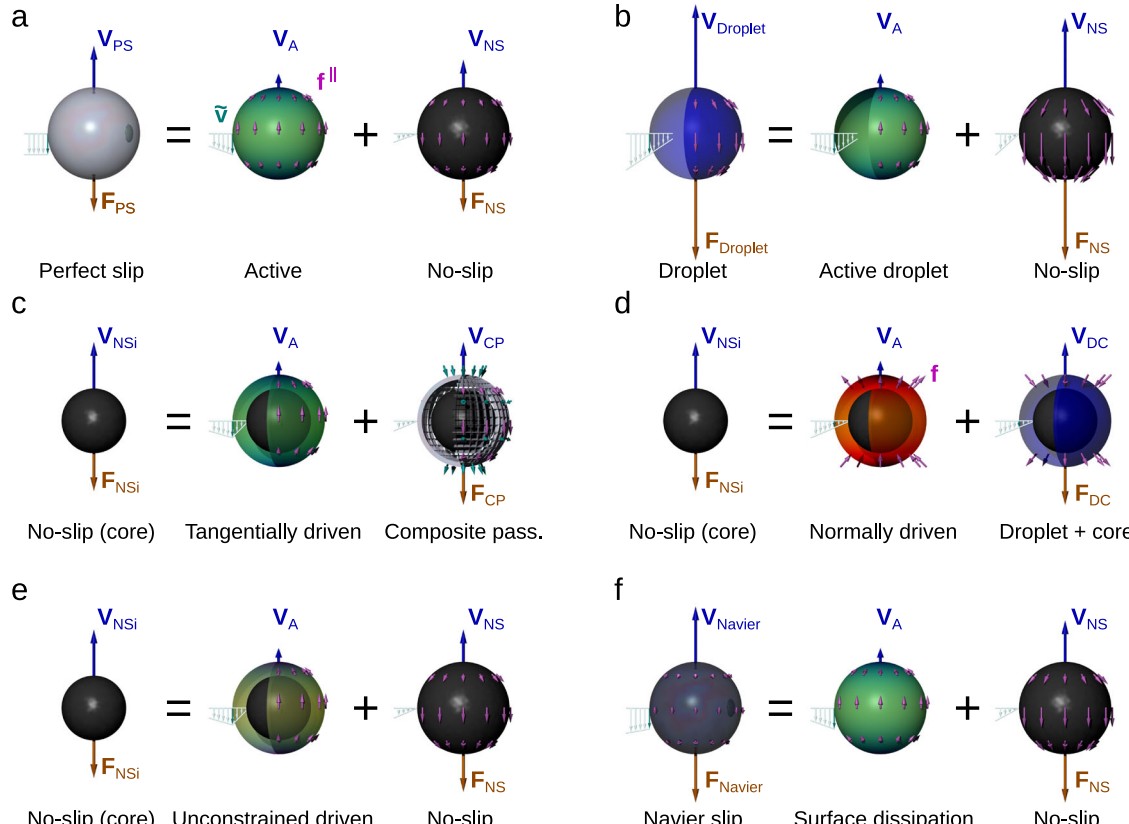

**Fig. 2 | Superposition principle used to derive different variants of the minimum dissipation theorem.** The passive body (*V*-problem, left) can be represented as a superposition of the optimal active swimmer (center) and another passive body (*F*-problem, right). The blue arrows indicate the velocity **V** (in the laboratory frame) of each body and the red arrows the drag force **F** on passive bodies, all drawn to scale. The cyan arrows indicate the velocity v̄ and its gradient in the co-moving frame (not to scale). **a** A perfect-slip body as a superposition between an optimal swimmer (external dissipation only) and a no-slip body, as shown in ref. 38.

**b** A droplet as a superposition between the optimal active droplet and a no-slip body. **c** A no-slip body as a superposition between a swimmer driven by tangential forces at the outer layer and a composite passive body. **d** A no-slip body as a superposition between a swimmer driven by normal forces and a droplet with a no-slip core. **e** A no-slip body as a superposition between a swimmer driven by unconstrained forces and a no-slip body at the outer surface. **f** A Navier-slip body as a superposition between a surface-driven swimmer with internal dissipation and a no-slip body.

a generic shape

$$P_A \geq \mathbf{V}_A \cdot \left( \mathbf{R}_V^{-1} - \mathbf{R}_F^{-1} \right)^{-1} \cdot \mathbf{V}_A. \tag{1}$$

The theorem expresses the minimum bound on the dissipation of the active swimmer $P_A$ with two passive hydrodynamic drag coefficients, represented by $6 \times 6$ matrices. $\mathbf{R}_V$ is the drag coefficient of a passive body with minimum dissipation while fulfilling the boundary conditions required by the active problem. $\mathbf{R}_F$ is the drag coefficient of another passive body that appears in the superposition and has a velocity distribution orthogonal to the active driving forces. The equality in the theorem (1) is satisfied exactly when the active flow represents a linear superposition of the two passive problems. We denote them as the *V*-problem and the *F*-problem because the velocity distribution of the optimal active swimmer is determined by the solution of the *V*-problem and the active forces by the solution of the *F*-problem. Besides giving the lower limit on dissipation, the theorem also solves the optimization problem by providing the distribution of velocities and active forces that allow the swimmer to achieve a given velocity with minimum dissipation.

We now provide a proof of the theorems stated above. The derivation of the minimum dissipation theorems is based on two crucial steps: first, we find the solution for the minimum dissipation for the motion of a body of a given shape, driven by an external force. In our previous work, which considered external dissipation only[38], this was

the perfect-slip body, with properties similar to those of an idealized air bubble in the fluid, as depicted in Fig. 2a. The second step that makes the problem solvable consists of finding another passive problem and form a linear superposition of its flow and that of the active body. Importantly, the dissipation in the superposition flow needs to be the sum of the dissipations of the two problems each on its own. Because dissipation is a quadratic function, the latter condition is not trivial and requires the application of the Lorentz reciprocal theorem[40]. We therefore first generalize the Helmholtz minimum dissipation theorem for passive bodies and prove that Stokes flows including fluid–fluid interfaces and surfaces with the Navier slip boundary condition take the form with minimum dissipation. We subsequently apply the principle of superposition between an active swimmer and a passive body to derive the theorems for all types of swimmers listed in Table 1.

### Generalization of passive minimum dissipation theorems

We first show that flows around passive bodies with several different boundary conditions (listed in Table 2) satisfy a minimum dissipation theorem, i.e., that any other flow satisfying less stringent boundary conditions has a higher dissipation rate. These are generalizations of the Helmholtz minimum dissipation theorem, which states that among all flows that satisfy the prescribed velocity boundary conditions and incompressibility, but not necessarily the Stokes equation, the actual Stokes flow has the smallest dissipation[41]. In the derivation of the theorem for external dissipation[38], we used the statement that among

**Table 2 | List of boundary conditions**

| Type | Velocity | Stress |
|---|---|---|
| No slip | $\mathbf{v} = \mathbf{0}$ | – |
| Perfect slip | $v^{\perp} = 0$ | $\mathbf{f}^{\parallel} = \mathbf{0}$ |
| Navier slip | $v^{\perp} = 0$ | $\mathbf{f}^{\parallel} = \frac{\mu}{\lambda}\mathbf{v}$ |
| Fluid–fluid interface | $v^{\perp} = v_{\mathrm{i}}^{\perp} = 0,\ \mathbf{v}^{\parallel} = \mathbf{v}_{\mathrm{i}}^{\parallel}$ | $\mathbf{f}^{\parallel} - \mathbf{f}_{\mathrm{i}}^{\parallel} = \mathbf{0}$ |
| No tangential slip shell | $\mathbf{v} = \mathbf{v}_{\mathrm{i}},\ \mathbf{v}^{\parallel} = \mathbf{0}$ | $f^{\perp} - f_{\mathrm{i}}^{\perp} = 0$ |

$\mathbf{f} = \boldsymbol{\sigma} \cdot \mathbf{n},\ \mathbf{f}_{\mathrm{i}} = \boldsymbol{\sigma}_{\mathrm{i}} \cdot \mathbf{n}.$

all bodies of a given shape that move with a given speed $V$ and have zero normal velocity on their surface (i.e., the fluid can not cross the body's surface), the perfect-slip body has the lowest dissipation. Here, we extend the theorem to two further types of boundary conditions.

First, we generalize the theorem to shape-preserving fluid–fluid interfaces. If the body has a fixed shape, but its interior contains another fluid such that the boundary condition in the co-moving frame reads $\tilde{v}^{\perp} = \tilde{v}_{\mathrm{i}}^{\perp} = 0$ and $\tilde{\mathbf{v}}^{\parallel} = \tilde{\mathbf{v}}_{\mathrm{i}}^{\parallel}$, then the minimum dissipation is reached when the tangential stress on the surface vanishes, $(\mathbf{I} - \mathbf{nn}) \cdot (\boldsymbol{\sigma} - \boldsymbol{\sigma}_{\mathrm{i}}) \cdot \mathbf{n} = \mathbf{0}$. The quantities labeled with the index i refer to the internal fluid domain and those without to the external. $\mathbf{n}$ denotes the outward pointing surface normal. The perpendicular and parallel components of the velocity are defined as $\tilde{v}^{\perp} = \mathbf{n} \cdot \tilde{\mathbf{v}}$ and $\tilde{\mathbf{v}}^{\parallel} = (\mathbf{I} - \mathbf{nn}) \cdot \tilde{\mathbf{v}}$, respectively. The theorem states that the dissipation of any flow around an interface with velocity continuity, itself moving with velocity $\mathbf{V}$, satisfies the inequality

$$P \geq \mathbf{V} \cdot \mathbf{R}_{\mathrm{Droplet}} \cdot \mathbf{V}. \tag{2}$$

Here, $\mathbf{R}_{\mathrm{Droplet}}$ denotes the generalized drag coefficient of the body described by the fluid-fluid interface, e.g., an oil droplet in water. A proof of the statement is given in Supplementary Note 1. A related version of the theorem has been derived in Ref. [42] for droplet suspensions. An immediate implication of Eq. (2) is that the dissipation of the flow around a droplet is always smaller than around a no-slip body of the same shape and the same velocity. Therefore, the matrix $\mathbf{R}_{\mathrm{NS}} - \mathbf{R}_{\mathrm{Droplet}}$ is always positive definite, which naturally also holds for the matrices $\mathbf{R}_{\mathrm{NS}}$ and $\mathbf{R}_{\mathrm{Droplet}}$. The expression $(\mathbf{R}_{\mathrm{Droplet}}^{-1} - \mathbf{R}_{\mathrm{NS}}^{-1})^{-1}$ can be rewritten as $\mathbf{R}_{\mathrm{Droplet}} \cdot (\mathbf{R}_{\mathrm{NS}} - \mathbf{R}_{\mathrm{Droplet}})^{-1} \cdot \mathbf{R}_{\mathrm{Droplet}} + \mathbf{R}_{\mathrm{Droplet}}$ and is therefore positive-definite as well. We have thus proven that the lower bound given by Eq. (1) is always positive.

In the second generalization, we introduce an energetic cost to the slip velocity on the surface, such that the total dissipation is given by (note the distinction between the velocity in the laboratory frame $\mathbf{v}$ and in the co-moving frame $\tilde{\mathbf{v}}$)

$$P = \int_{\mathcal{S}} \mathrm{d}S \left( -\mathbf{f} \cdot \mathbf{v} + \frac{\mu}{\lambda}\tilde{\mathbf{v}}^2 \right), \tag{3}$$

with the traction $\mathbf{f} = \boldsymbol{\sigma} \cdot \mathbf{n}$ defined as the force density exerted by the fluid on the body. Here the first term represents the power transferred from the swimmer to the fluid, which is identical to the total dissipation in the fluid (Supplementary Note 2). We therefore refer to it as external dissipation. The second term represents the internal dissipation, which is the cost of maintaining the velocity on the surface. We wrote the dissipation density, which can be arbitrary, as $\mu/\lambda$ in anticipation of the result that follows. The total dissipation as defined in Eq. (3) is minimal when the flow satisfies the Navier slip condition on the surface

$$\tilde{\mathbf{v}} = \frac{\lambda}{\mu} \mathbf{f}^{\parallel}. \tag{4}$$

Here $\lambda$, which we initially introduced as a free parameter, takes the role of the slip length and $\mu$ denotes the viscosity. We note that $\lambda = 0$ for the

no-slip condition and $\lambda = \infty$ for the perfect slip. The total dissipation in any flow around that body moving with velocity $\mathbf{V}$ satisfies the inequality

$$P \geq \mathbf{V} \cdot \mathbf{R}_{\mathrm{Navier}} \cdot \mathbf{V} \tag{5}$$

where $\mathbf{R}_{\mathrm{Navier}}$ denotes the generalized drag coefficient of the rigid body with shape $\mathcal{S}$ and slip length $\lambda$. The statement is proven in Supplementary Note 1. By inserting the no-slip flow into the inequality, we see that $\mathbf{R}_{\mathrm{NS}} - \mathbf{R}_{\mathrm{Navier}}$ is positive definite and the lower bound in Eq. (1) is positive.

**Derivation of active minimum dissipation theorems**
In the second step, we will derive active minimum dissipation theorems for 5 swimmer types that include internal dissipation, as listed in Table 1. For that purpose, we will apply the derived inequalities to the superposition of an active swimmer and a passive body of the same shape. The crucial step is always to find two passive problems that satisfy the boundary conditions of the active problem whereby the first problem possesses minimum dissipation and the second problem has a velocity distribution orthogonal to the forces in the active problem.

**Active surface-driven droplet.** We start by deriving the theorem for the active droplet, which is propelled by an active tangential force on the surface (Fig. 1a). The latter determines the stress discontinuity, $(\mathbf{I} - \mathbf{nn}) \cdot (\boldsymbol{\sigma} - \boldsymbol{\sigma}_{\mathrm{i}}) \cdot \mathbf{n} = \mathbf{f}_{\mathrm{A}}^{\parallel}$. Such active stress can result from the Marangoni effect, where it is caused by a gradient in the surface tension and is widely used to propel active droplets[43]. The active power exerted by this force is

$$P_{\mathrm{A}} = -\int_{\mathcal{S}} \mathrm{d}S\,\mathbf{f}_{\mathrm{A}} \cdot \mathbf{v}_{\mathrm{A}}. \tag{6}$$

To derive the minimum dissipation theorem for the active droplet, we need one passive body that minimizes the dissipation while fulfilling the boundary condition on the surface and another passive body with a velocity distribution that is orthogonal to the active forces (see below) while also satisfying the boundary condition. The former is represented by a droplet and the latter by a no-slip body. We first calculate the dissipation in a flow that is a linear superposition between the active swimmer, moving with velocity $\mathbf{V}_{\mathrm{A}}$, and a passive hollow no-slip body, moving with velocity $\mathbf{V}_{\mathrm{NS}}$, as illustrated in Fig. 2b. The dissipated power in the superposition flow can be expressed as

$$P_{\mathrm{A+NS}} = -\int_{\mathcal{S}} \mathrm{d}S\,(\mathbf{f}_{\mathrm{A}} + \mathbf{f}_{\mathrm{NS}}) \cdot (\mathbf{v}_{\mathrm{A}} + \mathbf{v}_{\mathrm{NS}}). \tag{7}$$

We now apply the Lorentz reciprocal theorem (see Methods) by integrating over the whole space, without surface contributions. However, because the traction is concentrated on the surface $\mathcal{S}$, we write its contribution in the form of a surface integral. From Eq. (45) it follows that the two mixed terms are identical $\int_{\mathcal{S}} \mathrm{d}S\,\mathbf{f}_{\mathrm{NS}} \cdot \mathbf{v}_{\mathrm{A}} = \int_{\mathcal{S}} \mathrm{d}S\,\mathbf{f}_{\mathrm{A}} \cdot \mathbf{v}_{\mathrm{NS}}$. By expressing the velocities in the co-moving system as in Eq. (46), we also know that $\int_{\mathcal{S}} \mathrm{d}S\,\mathbf{f}_{\mathrm{A}} \cdot \mathbf{v}_{\mathrm{NS}} = \int_{\mathcal{S}} \mathrm{d}S\,\mathbf{f}_{\mathrm{A}} \cdot \tilde{\mathbf{v}}_{\mathrm{NS}} + \mathbf{F}_{\mathrm{A}} \cdot \mathbf{V}_{\mathrm{NS}} = 0$. The latter follows from $\tilde{\mathbf{v}}_{\mathrm{NS}} = 0$ and $\mathbf{F}_{\mathrm{A}} = 0$. Therefore, the mixed terms vanish and Eq. (7) reduces to

$$P_{\mathrm{A+NS}} = P_{\mathrm{A}} - \mathbf{F}_{\mathrm{NS}} \cdot \mathbf{V}_{\mathrm{NS}} \tag{8}$$

i.e., the sum of the dissipation contributions by the active swimmer and the no-slip body, each on its own. We have thus *a posteriori* justified the choice of the problem used in the superposition. This decomposition is a crucial step that is decisive for the feasibility of the solution.

Finally, we know that like any flow satisfying the boundary conditions of continuous tangential and zero normal velocity on the

surface, the dissipation of the superposition flow satisfies the inequality (2), which states that $P_{A+NS} \geq (\mathbf{V}_A + \mathbf{V}_{NS}) \cdot \mathbf{R}_{Droplet} \cdot (\mathbf{V}_A + \mathbf{V}_{NS})$. The equality is fulfilled exactly when the superposition corresponds to the flow around a droplet. This implies $\mathbf{V}_A + \mathbf{V}_{NS} = \mathbf{V}_{Droplet}$ and $\mathbf{F}_{NS} = \mathbf{F}_{Droplet}$, along with $\mathbf{F}_{Droplet} = \mathbf{R}_{Droplet} \cdot \mathbf{V}_{Droplet}$ and $\mathbf{F}_{NS} = \mathbf{R}_{NS} \cdot \mathbf{V}_{NS}$. These equations are solved to yield

$$\mathbf{V}_{Droplet} = \left( \mathbf{I} - \mathbf{R}_{NS}^{-1} \cdot \mathbf{R}_{Droplet} \right)^{-1} \cdot \mathbf{V}_A, \qquad (9)$$

$$\mathbf{V}_{NS} = \left( \mathbf{R}_{Droplet}^{-1} \cdot \mathbf{R}_{NS} - \mathbf{I} \right)^{-1} \cdot \mathbf{V}_A. \qquad (10)$$

With these velocities, we finally obtain the minimum dissipation theorem

$$P_A \geq \mathbf{V}_A \cdot \left( \mathbf{R}_{Droplet}^{-1} - \mathbf{R}_{NS}^{-1} \right)^{-1} \cdot \mathbf{V}_A. \qquad (11)$$

Because the equality in the theorem is fulfilled exactly if the superposition gives the flow around a droplet, we also know the distribution of velocities and active forces that minimize the dissipation while maintaining the swimming speed $\mathbf{V}_A$. The optimal slip velocity is given by the flow around the droplet $\tilde{\mathbf{v}}_A = \tilde{\mathbf{v}}_{Droplet}$ and the tangential traction is opposite-equal to that of a no-slip body $\mathbf{f}_A^{\parallel} = -\mathbf{f}_{NS}^{\parallel}$.

As a simple example, we apply the theorem to a spherical viscous droplet with external viscosity $\mu$, internal viscosity $\mu_i$ and radius $a$. According to the Hadamard–Rybczynski equation, the drag coefficient of the droplet is[41]

$$R_{Droplet} = 6\pi\mu a \cdot \frac{\mu_i + \frac{2}{3}\mu}{\mu_i + \mu} \qquad (12)$$

and we obtain

$$P_A \geq 6\pi(2\mu + 3\mu_i) a V_A^2. \qquad (13)$$

For equal viscosities, $\mu_i = \mu$, the ratio of internal vs. external dissipation is $3:2$. The minimum dissipation then becomes $30\pi\mu a V_A^2$. The simple additivity of external and internal dissipation only holds for a sphere. In general, the presence of internal dissipation will influence the optimal external velocity profile and vice versa.

**Swimmer with an extended propulsive layer and tangential forces.**
The second problem we solve involves a swimmer with a no-slip surface $\mathcal{S}_i$ that is propelled by tangential forces on a closed outer surface $\mathcal{S}$ (Fig. 1b), also called control surface[25]. Such a model has been proposed to describe the ciliary layer, for example in *Volvox*[44]. We now apply the inequality for a no-slip body with the shape of the inner surface ($\mathcal{S}_i$). The superposition is formed with a composite passive body that has a no-slip boundary $\tilde{\mathbf{v}}_{CP} = \mathbf{0}$ at $\mathcal{S}_i$ (Fig. 2c). At the outer surface ($\mathcal{S}$), the boundary condition is zero tangential velocity $(\mathbf{I} - \mathbf{nn}) \cdot \tilde{\mathbf{v}}_{CP} = \mathbf{0}$ and continuity of normal stress $\mathbf{n} \cdot \boldsymbol{\sigma} \cdot \mathbf{n}$, i.e., zero normal traction $\mathbf{n} \cdot \mathbf{f}_{CP} = 0$. The definition of the composite body ensures the orthogonality between active forces and the passive flow, which is required in the derivation below. At the same time, the choice of tangential-only forces at the outer surface ensures that it is always possible to find an active swimmer that exactly satisfies the superposition condition.

We can express the dissipation in the superposition flow as

$$P_{A+CP} = -\int_{\mathcal{S}+\mathcal{S}_i} dS (\mathbf{f}_A + \mathbf{f}_{CP}) \cdot (\mathbf{v}_A + \mathbf{v}_{CP}). \qquad (14)$$

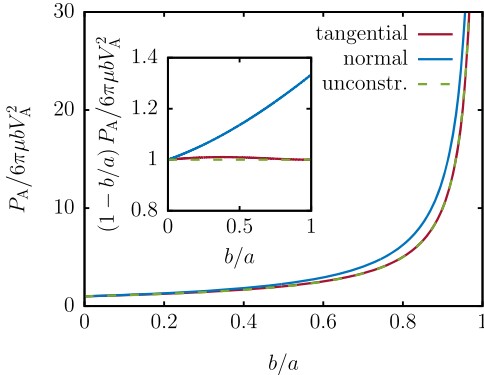

**Fig. 3 | Dissipation by spherical swimmers with an extended propulsive layer.** Lower bound on dissipation by a spherical swimmer (radius $b$) with an extended propulsive layer, such that the active forces act on a concentric sphere with radius $a$. The red solid lines show the case of tangential propulsive forces (18), the blue solid lines the case of normal forces (22) and the green dashed lines the case of unconstrained forces (26).

We now employ the Lorentz reciprocal theorem on the fluid domain outside $\mathcal{S}_i$ with an additional surface integral of the traction on $\mathcal{S}$. By combining the forms (45) and (46) we obtain $\int_{\mathcal{S}+\mathcal{S}_i} dS \, \mathbf{f}_{CP} \cdot \mathbf{v}_A = \int_{\mathcal{S}+\mathcal{S}_i} dS \, \mathbf{f}_A \cdot \mathbf{v}_{CP} = \int_{\mathcal{S}+\mathcal{S}_i} dS \, \mathbf{f}_A \cdot \tilde{\mathbf{v}}_{CP} + \mathbf{F}_A \cdot \mathbf{V}_{CP} = 0$. On the outer surface, the traction $\mathbf{f}_A$ is by definition tangential and $\tilde{\mathbf{v}}_{CP}$ normal to the surface, therefore their product is zero. This also holds for the inner surface where $\tilde{\mathbf{v}}_{CP} = 0$. Together with $\mathbf{F}_A = 0$, all terms are identically zero. Therefore, the mixed terms in Eq. (14) integrate to zero and we have proven the additivity of the dissipated power

$$P_{A+CP} = P_A - \mathbf{F}_{CP} \cdot \mathbf{V}_{CP}. \qquad (15)$$

From here, the new version of the minimum dissipation theorem follows in complete analogy to the derivation of Eq. (11):

$$P_A \geq \mathbf{V}_A \cdot \left( \mathbf{R}_{NSi}^{-1} - \mathbf{R}_{CP}^{-1} \right)^{-1} \cdot \mathbf{V}_A \qquad (16)$$

where $\mathbf{R}_{NSi}$ is the drag coefficient of a no-slip body described by the inner surface $\mathcal{S}_i$, and $\mathbf{R}_{CP}$ is the drag coefficient of a composite passive body imposing a no-slip boundary condition at $\mathcal{S}_i$, with zero tangential velocity and zero normal traction at $\mathcal{S}$. At the surface $\mathcal{S}$ we have $\mathbf{f}_{NSi}$ and $\tilde{\mathbf{v}}_{CP}^{\parallel} = 0$. The optimal swimmer therefore has a propulsive force $\mathbf{f}_A = -\mathbf{f}_{CP}$ and velocity $\tilde{\mathbf{v}}_A^{\parallel} = \tilde{\mathbf{v}}_{NSi}^{\parallel}$ at the outer surface.

For a spherical body, the drag coefficient of the composite passive body with the outer radius $a$ and the inner radius $b$ is (Supplementary Note 3)

$$R_{CP} = 8\pi\mu \cdot \frac{20a^5 + 11a^4 b + 11a^3 b^2 + a^2 b^3 + ab^4 + b^5}{28a^4 + 13a^3 b + 13a^2 b^2 + 3ab^3 + 3b^4}. \qquad (17)$$

Together with the drag coefficient of the no-slip body $R_{NS} = 6\pi\mu b$, we obtain the limit on dissipation

$$P_A \geq 24\pi\mu b V_A^2 \cdot \frac{20a^5 + 11a^4 b + 11a^3 b^2 + a^2 b^3 + ab^4 + b^5}{5(a-b)\left(4a^2 + ab + b^2\right)^2}. \qquad (18)$$

The dissipation as a function of the ratio of the inner to outer radius $b/a$ is shown in Fig. 3. If the propulsive layer is thick compared to the particle size, $a \gg b$, the dissipation bound is $6\pi\mu b V_A^2$, which is the dissipation by an externally driven sphere. This is also the lowest possible dissipation in a flow around a no-slip sphere. In the limit $b \to a$,

if the propulsive layer becomes thin, the lowest dissipation is $6\pi\mu V_A^2 b/(a-b)$.

**Swimmer propelled by normal forces.** An additional problem that can be solved is the dissipation by a swimmer that is also propelled by forces acting at the outer surface ($\mathcal{S}$), but this time the active forces have a direction normal to the surface, i.e., $\mathbf{f}_A \parallel \mathbf{n}$ (Fig. 1c). The motivation originates from phoretic swimmers, where the fluid is set in motion by a potential (normal force) gradient in the boundary layer. Other boundary conditions are $\tilde{\mathbf{v}}_A = 0$ at the inner surface $\mathcal{S}_i$ and the continuity of velocity at the outer surface.

As in the previous case, we apply the inequality to a body with a no-slip boundary and the shape of the inner surface $\mathcal{S}_i$. We form the superposition between the active swimmer and a passive body consisting of a no-slip core at $\mathcal{S}_i$ and a fluid–fluid interface at $\mathcal{S}$ (Fig. 2d). Again, we verify the additivity of dissipation for this superposition, which can be expressed as

$$P_{A+DC} = -\int_{\mathcal{S}+\mathcal{S}_i} dS\,(\mathbf{f}_A + \mathbf{f}_{DC})\cdot(\mathbf{v}_A + \mathbf{v}_{DC}). \tag{19}$$

Again we apply the Lorentz reciprocal theorem (Eqs. (45) and (46)) on the fluid domain outside $\mathcal{S}_i$ with additional tractions on $\mathcal{S}$ to show $\int_{\mathcal{S}+\mathcal{S}_i} dS\,\mathbf{f}_{DC}\cdot\mathbf{v}_A = \int_{\mathcal{S}+\mathcal{S}_i} dS\,\mathbf{f}_A\cdot\mathbf{v}_{DC} = \int_{\mathcal{S}+\mathcal{S}_i} dS\,\mathbf{f}_A\cdot\tilde{\mathbf{v}}_{DC} + \mathbf{F}_A\cdot\mathbf{V}_{DC} = 0$. Again, we have $\mathbf{f}_A\cdot\tilde{\mathbf{v}}_{DC} = 0$ at both integration surfaces. At the outer surface $\mathcal{S}$ this is because the traction $\mathbf{f}_A$ is normal to the surface and the velocity $\tilde{\mathbf{v}}_{DC}$ is tangential. At the inner surface $\mathcal{S}_i$ their product also vanishes because $\tilde{\mathbf{v}}_{DC} = 0$. Therefore, the integrals of the mixed terms in Eq. (19) vanish, proving the additivity $P_{A+DC} = P_A - \mathbf{F}_{DC}\cdot\mathbf{V}_{DC}$. The corresponding minimum dissipation theorem reads

$$P_A \geq \mathbf{V}_A\cdot\left(\mathbf{R}_{NSi}^{-1} - \mathbf{R}_{DC}^{-1}\right)^{-1}\cdot\mathbf{V}_A. \tag{20}$$

The superposition also states that the optimal swimmer has the active forces $\mathbf{f}_A = -\mathbf{f}_{DC}$ and normal velocity $\tilde{\mathbf{v}}_A^\perp = \tilde{\mathbf{v}}_{NSi}^\perp$ at the outer surface.

As an example, we calculate the dissipation by a spherical swimmer with inner radius $b$ and radius of the propulsive layer $a$. The drag coefficient of the droplet with a no-slip core is (Supplementary Note 3)

$$R_{DC} = 8\pi\mu b\,\frac{5a^3 + 6a^2 b + 3ab^2 + b^3}{8a^2 b + 9ab^2 + 3b^3} \tag{21}$$

Together with the $R_{NS} = 6\pi\mu b$, we obtain

$$P_A \geq 6\pi\mu b V_A^2\cdot\frac{4(5a^3 + 6a^2 b + 3ab^2 + b^3)}{5(2a+b)^2(a-b)}. \tag{22}$$

The above dependence is shown by the blue line in Fig. 3. In the limit $b \to 0$, the lower bound is $6\pi\mu b V_A^2$, which corresponds to a sphere pulled by an external force. For $b \to a$, the result is $8\pi\mu b^2/(a-b)$. While the expression is similar as for tangential propulsion, the dissipation is higher by a factor of 4/3.

**Swimmer propelled by unconstrained forces.** We now relax the constraint from the last two cases where we required the forces to be tangential or normal and allow any distribution of forces acting at the outer surface ($\mathcal{S}$) (Fig. 1d). As illustrated in Fig. 2e, we use a superposition between the active swimmer and a no-slip body described by the outer surface and apply the inequality to the inner shape.

The dissipation of the superposition now reads

$$P_{A+NS} = -\int_{\mathcal{S}+\mathcal{S}_i} dS\,(\mathbf{f}_A + \mathbf{f}_{NS})\cdot(\mathbf{v}_A + \mathbf{v}_{NS}). \tag{23}$$

We apply the Lorentz reciprocal theorem (Eqs. (45) and (46)) on the volume outside the surface $\mathcal{S}_i$ with additional traction at $\mathcal{S}$. Because the velocity field is also needed in the space between the to surfaces, we treat the no-slip body as a hollow fluid-filled shell such that $\tilde{\mathbf{v}}_{NS} = 0$ inside $\mathcal{S}$. We again see that the mixed terms disappear because $\tilde{\mathbf{v}}_{NS} = 0$ and show the additivity

$$P_{A+NS} = P_A - \mathbf{F}_{NS}\cdot\mathbf{V}_{NS}. \tag{24}$$

Now the inequality reads $P_{A+NS} \geq (\mathbf{V}_A + \mathbf{V}_{NS})\cdot\mathbf{R}_{NSi}\cdot(\mathbf{V}_A + \mathbf{V}_{NS})$. Combined, these two equations give the minimum dissipation theorem

$$P_A \geq \mathbf{V}_A\cdot\left(\mathbf{R}_{NSi}^{-1} - \mathbf{R}_{NS}^{-1}\right)^{-1}\cdot\mathbf{V}_A \tag{25}$$

where $\mathbf{R}_{NS}$ and $\mathbf{R}_{NSi}$ are the drag coefficients of no-slip bodies with shapes $\mathcal{S}$ and $\mathcal{S}_i$, respectively. The active traction on the outer surface of the optimal swimmer is $\mathbf{f}_A = -\mathbf{f}_{NS}$ and the velocity $\tilde{\mathbf{v}}_A = \tilde{\mathbf{v}}_{NSi}$.

In the case of a spherical swimmer with $R_{NS} = 6\pi\mu a$ and $R_{NSi} = 6\pi\mu b$, the minimum dissipation is

$$P_A \geq 6\pi\mu b V_A^2\cdot\frac{a}{a-b} \tag{26}$$

which is naturally always lower than the dissipation in the more restrictive cases of tangential or normal forces (Fig. 3). The advantage over tangential propulsion is small, however, and never exceeds $\lesssim 1\%$ for spherical swimmers.

**Swimmer with surface dissipation.** The last problem for which we can calculate the efficiency limit is a surface-driven swimmer for which the maintenance of the slip velocity comes with an energetic cost, namely with the power density $\zeta_s \tilde{\mathbf{v}}^2$, where $\zeta_s$ is a constant that characterizes the efficiency of the propulsion mechanism (Fig. 1e). The total dissipated power is defined as

$$P_A = \int_{\mathcal{S}} dS\left(-\mathbf{f}_A\cdot\mathbf{v}_A + \zeta_s \tilde{\mathbf{v}}_A^2\right). \tag{27}$$

In this case, we form the superposition with a no-slip body of the same shape and derive an expression for the dissipation of the superposition flow, including internal dissipation (Fig. 2f),

$$P_{A+NS} = \int_{\mathcal{S}} dS\left(-(\mathbf{f}_A + \mathbf{f}_{NS})\cdot(\mathbf{v}_A + \mathbf{v}_{NS}) + \zeta_s \tilde{\mathbf{v}}_A^2\right). \tag{28}$$

In the second term, we took into account that $\tilde{\mathbf{v}}_{NS} = 0$ and therefore only $\tilde{\mathbf{v}}_A$ contributes to the internal dissipation. From the reciprocal theorem (Eqs. (45) and (46)) it follows that $\int_{\mathcal{S}} dS\,\mathbf{f}_{NS}\cdot\mathbf{v}_A = \int_{\mathcal{S}} dS\,\mathbf{f}_A\cdot\mathbf{v}_{NS} = \int_{\mathcal{S}} dS\,\mathbf{f}_A\cdot\tilde{\mathbf{v}}_{NS} + \mathbf{F}_A\cdot\mathbf{V}_{NS} = 0$, proving the additivity

$$P_{A+NS} = P_A - \mathbf{F}_{NS}\cdot\mathbf{V}_{NS}. \tag{29}$$

The choice of the $V$-problem here differs from the previous cases where it only needed to fulfill the boundary condition, i.e., zero normal velocity on the surface. Here, we need a problem that minimizes the dissipation including the internal contribution. This is the case for a body with the Navier slip condition on the surface. Therefore, the superposition flow satisfies the inequality (5)

$$P_{A+NS} \geq (\mathbf{V}_A + \mathbf{V}_{NS})\cdot\mathbf{R}_{Navier}\cdot(\mathbf{V}_A + \mathbf{V}_{NS}). \tag{30}$$

In analogy with the previous cases, we derive the minimum dissipation theorem which reads

$$P_A \geq \mathbf{V}_A \cdot \left( \mathbf{R}_{\text{Navier}}^{-1} - \mathbf{R}_{\text{NS}}^{-1} \right)^{-1} \cdot \mathbf{V}_A, \tag{31}$$

where $\mathbf{R}_{\text{Navier}}$ is the drag coefficient of a body with the Navier-slip boundary with slip length $\lambda = \mu/\zeta_s$. In the limit $\zeta_s \to 0$, we obtain the perfect slip body, as in the original theorem for external-only dissipation[38].

As in the other cases, the superposition states that the slip velocity of the optimal swimmer is $\tilde{\mathbf{v}}_A = \tilde{\mathbf{v}}_{\text{Navier}}$. However, the internal dissipation model differs and presents an exception with regard to the optimal active forces. Among the cases discussed here, it is the only one where both passive problems that form the superposition have non-zero tractions. Therefore, the optimal swimmer has the traction $\mathbf{f}_A^{\parallel} = \mathbf{f}_{\text{Navier}}^{\parallel} - \mathbf{f}_{\text{NS}}^{\parallel}$.

In the simplest case of a spherical swimmer, the drag coefficient of the sphere with Navier slip boundary is[45]

$$R_{\text{Navier}} = 6\pi\mu a \cdot \frac{1 + 2\lambda/a}{1 + 3\lambda/a}. \tag{32}$$

The resulting active dissipation is

$$P_A \geq 6\pi\mu a V_A^2 \left( 2 + \frac{a}{\lambda} \right) = 6\pi a V_A^2 (2\mu + a\zeta_s). \tag{33}$$

Again, the additive nature of the external and internal contributions is limited to spherical geometry.

In the limit $\lambda \ll a$, the leading term in the dissipation agrees with that with external propulsion if we use $a$ for the inner radius and $a + \lambda$ for the outer.

## Swimmers with optimal shapes

The minimum dissipation theorem provides us with the lowest value of dissipation rate by a swimmer of a given shape, moving with a given velocity. We can now go one step further and ask the question about the lowest possible dissipation by a swimmer of a given volume, regardless of its shape. In the following, we apply the newly derived theorems to determine the optimal shapes of microswimmers with combined internal and external dissipation. The optimization problem consists of minimizing the dissipation while keeping the volume and the swimming speed constant. It is well known that the optimal shape with external dissipation only is not well defined because the dissipation vanishes in the limit of an infinitely elongated needle[34]. In the opposite limit, when the internal dissipation, modeled with the term $\zeta_s \tilde{\mathbf{v}}^2$, dominates, the optimal swimmer shape consists of a body with two thin elongated protrusions along the symmetry axis[24]. Also, those shapes could only be determined by additionally restricting the allowed curvature of the surface with a dimensionless minimum radius $\hat{r}_{\min}$, normalized such that $\hat{r}_{\min} = 1$ restricts the shape to a spherical body.

**Optimal shape of the active droplet.** In the following, we numerically determine the optimal shape of a fluid-filled surface-driven swimmer of the type of a self-propelled droplet. For each axisymmetric shape, we numerically determine the drag coefficients of the passive droplet and the no-slip body using a custom-written boundary element method (based on Green's functions from BEMLIB[46]) and use the theorem to calculate the minimum dissipation by the active droplet. We parameterize the shape as a chain of segments of equal length, linearly rescale it to impose a fixed volume, and then use a numerical optimization procedure to determine the shape with the minimum dissipation.

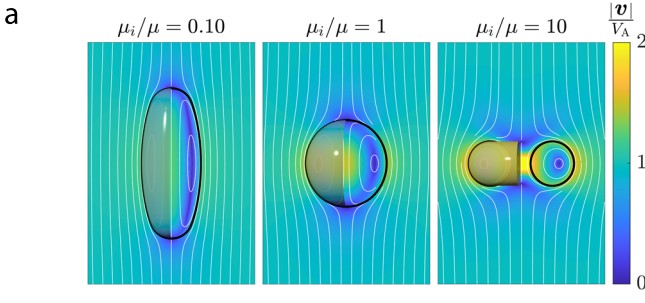

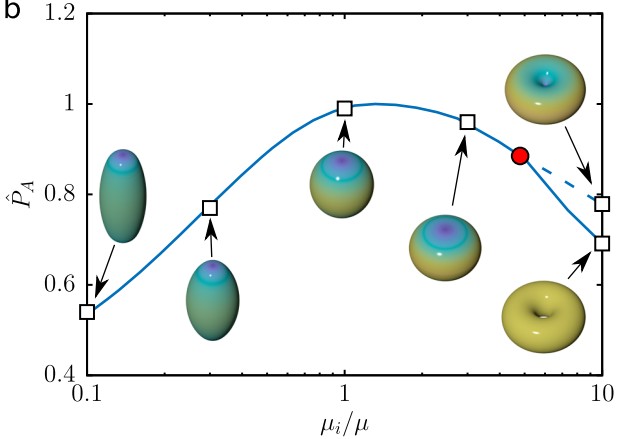

**Fig. 4 | Optimal shapes of active surface-driven droplets. a** The optimal shapes of active droplets with three different ratios of the internal to external viscosity $\mu_i/\mu$ and their flow fields. **b** The dissipation $\hat{P}_A = P_A/(6\pi(2\mu + 3\mu_i)aV_A^2)$ of the optimal active surface-driven droplet, relative to the equivalent droplet of a spherical shape (13), as a function of the viscosity ratio. The red dot indicates a topological bifurcation above which a toroidal shape (solid line) has a dissipation lower than the optimal swimmer with spherical topology (dashed line). The insets show the optimal shapes at viscosity ratios marked with squares.

The resulting shapes obtained for different ratios of the internal to external viscosity ($\mu_i/\mu$) are shown in Fig. 4a. Figure 4b shows the obtained minimum dissipation, relative to that of a spherical shape (13), along with a selection of optimal shapes. As anticipated, with a vanishing internal viscosity, $\mu_i = 0$, only external dissipation remains relevant and the optimal shape becomes that of an infinitely thin needle. In the case of equal viscosities, $\mu_i = \mu$, the optimal shape is close to a prolate spheroid with an aspect ratio of 1.16. The dissipated power is 0.991 that of a spherical droplet, indicating that the advantage over a spherical shape is tiny. On the other hand, if the internal viscosity dominates, the optimal shapes first become oblate and eventually transition to a toroid. The discontinuous topological transition takes place at a viscosity ratio $\mu_i/\mu \approx 4$. The propulsion by a toroid rotating inside-out has been studied in the literature for a long time[11,47,48], but only recently a flagellate with a swimming mode using the same principle has been reported[49].

**Optimal shape of the swimmer with surface dissipation.** The second class of swimmers for which we determine the optimal shapes are the swimmers with internal surface dissipation. Here we face the problem that the mathematically optimal shapes contain infinitely long axial protrusions. In order to perform the optimization among realistic shapes, we need to additionally restrict the radius of curvature. For any point on the surface, we demand that both principal curvatures $\kappa_{1,2} \leq 1/r_{\min}$, where the minimum radius is determined by its dimensionless value as $r_{\min} = \hat{r}_{\min} a$, where $a = \sqrt[3]{3V/4\pi}$ is the radius of the sphere with the equivalent volume. Again, we use a custom-written axisymmetric boundary element solver to determine the drag coefficients for a given shape (first rescaled to unit volume) and run

the shape parametrization through a constrained minimization routine.

Examples of optimal shapes for different internal dissipation densities $\zeta_s$ and curvature radii $\hat{r}_{min}$ are shown in Fig. 5a, with some of the flows shown in Fig. 5b. The minimum dissipated power $P_A$, scaled by the minimum dissipation of a spherical swimmer (33), as a function of both parameters is shown in Fig. 5c. We have previously shown that in the case of dominant internal dissipation[24], the advantage over a spherical swimmer is $\lesssim$20% over a wide range of realistic shapes. Here, we show that the optimal shape has a much larger influence on dissipation when the combination of internal and external dissipation is taken into account.

## Swimmer under external force

In addition to active propulsion, the swimmer can be subject to an external force, for example when moving in the presence of gravity. Here, we will determine how velocity and dissipation will be affected in the presence of an external force $\mathbf{F}_{ext}$. The assumption we make is that the active driving force density $\mathbf{f}_A$ remains constant, whereas the velocity and the dissipated power are affected by the external force. For example, in the case of the active droplet, we assume that the tangential driving force on the surface remains constant, but the surface velocity $\bar{\mathbf{v}}_A$ is reduced when the swimmer is pulling (or pushing) against a resistive load. The assumption of a fixed driving force is less straightforward in the case of internal dissipation, where we assume that the additional load affects the slip velocity in the same way as it would on a boundary with the Navier slip and see that the expression derived above remains valid.

Because the solution with an external, but without active forces, represents exactly the V-problem, the swimmer under load can be represented as a superposition of the unloaded active swimmer and the body from the V-problem (e.g., passive droplet or no-slip core). The velocity response to the applied force is therefore determined as

$$\mathbf{V}_A = \mathbf{V}_A^0 + \mathbf{R}_V^{-1} \cdot \mathbf{F}_{ext}. \tag{34}$$

Here, $\mathbf{V}_A^0$ is the unperturbed swimming velocity and $\mathbf{R}_V$ is the drag coefficient of the V-problem. Thus, the mobility of the passive body from the V-problem ($\mathbf{R}_V^{-1}$) acts as the velocity response function of the active swimmer.

Under the same assumption, the dissipated power by the active swimmer, which consists of the power produced by the swimmer and the power contributed by the external force, is (see Supplementary Note 4 for a derivation)

$$P_A(\mathbf{F}_{ext}) = P_A^0 + \mathbf{F}_{ext} \cdot \left( \mathbf{V}_A^0 + \mathbf{V}_A \right) \tag{35}$$

with $P_A^0 = P_A(\mathbf{F}_{ext} = 0)$ denoting the dissipated power in the absence of external force (Fig. 6). Interestingly, the optimal force distribution does not change with the applied load: the force distribution that minimizes dissipation when moving freely with a given velocity will still be optimal under load and always follows the force distribution from the F-problem. By expressing $\mathbf{F}_{ext}$ with $\mathbf{V}_A$ (34), the dissipation obtains the form

$$P_A(\mathbf{V}_A) = P_A^0 - \mathbf{V}_A^0 \cdot \mathbf{R}_V \cdot \mathbf{V}_A^0 + \mathbf{V}_A \cdot \mathbf{R}_V \cdot \mathbf{V}_A = P_A(\mathbf{V}_A = 0) + \mathbf{V}_A \cdot \mathbf{R}_V \cdot \mathbf{V}_A. \tag{36}$$

The second term is the dissipation that is expected for an object driven by an external force (Fig. 6). The first term represents the dissipation rate of a swimmer that is brought to a stall by an external force. Its lower bound is

$$P_A(\mathbf{V}_A = 0) \geq \mathbf{V}_A^0 \cdot \left[ \left( \mathbf{R}_V^{-1} - \mathbf{R}_F^{-1} \right)^{-1} - \mathbf{R}_V \right] \cdot \mathbf{V}_A^0. \tag{37}$$

In other words, the dissipation can be decomposed into one component that is required to run the force-generation mechanism $P_A(V_A = 0)$ and another one that is identical to the dissipation by a passive body from the V-problem (i.e., a perfect-slip body or a passive bubble).

## Rate of entropy production

By taking into account the thermal noise acting on the swimmer as well as an additional force $\mathbf{F}$, the equations of motion are

$$\dot{\mathbf{X}} = \left[ \mathbf{V}_A^0 + \mathbf{R}_V^{-1} \cdot \mathbf{F} + \boldsymbol{\xi} \right]_{[1,2,3]} \tag{38}$$

$$\dot{\mathbf{n}}_\alpha = \left[ \mathbf{V}_A^0 + \mathbf{R}_V^{-1} \cdot \mathbf{F} + \boldsymbol{\xi} \right]_{[4,5,6]} \times \mathbf{n}_\alpha \tag{39}$$

where $\mathbf{X}$ is the spatial coordinate of the particle and $\mathbf{n}_\alpha$ one of the three vectors describing the orientation of the swimmer. The product in the expression for rotational motion is carried out using the Stratonovich interpretation. The Brownian noise is characterized by its

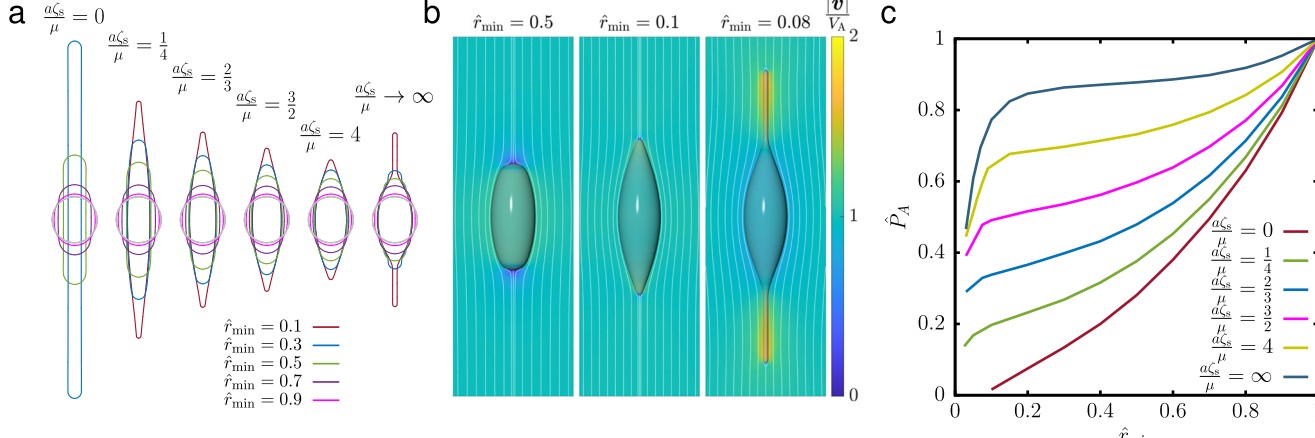

**Fig. 5 | Optimal shapes of swimmers with surface dissipation. a** Numerically obtained optimal shape that minimizes the dissipation as a function of the internal dissipation density $\zeta_s$ while keeping the total volume of the swimmer fixed ($V = 4\pi a^3/3$). The shapes are restricted by the minimum curvature radius $r_{min} = \hat{r}_{min} a$. **b** Streamlines in the co-moving frame and propulsion velocity $\bar{v}$ (color coded) for three optimal shapes, obtained with $a\zeta_s/\mu = 4$. **c** The dissipation by optimal swimmers as a function of the prescribed minimum curvature radius for a set of internal dissipation densities $a\zeta_s/\mu$. The dissipation is normalized by that of a spherical swimmer, $\hat{P}_A = P_A/(6\pi a V_A^2(2\mu + a\zeta_s))$.

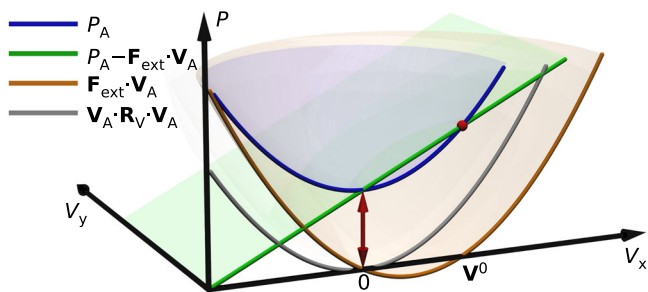

**Fig. 6 | Force-dependent dissipation.** Dissipation by an active microswimmer subject to an external force $\mathbf{F}_{\text{ext}}$. The dissipated power ($P_A$, blue) can be split into the power output of the swimmer ($P_A - \mathbf{F}_{\text{ext}} \cdot \mathbf{V}_A$, green) and the rate of work by the external force (brown). The dissipated power differs from the expression $\mathbf{V}_A \cdot \mathbf{R}_V \cdot \mathbf{V}_A$ (gray) by a constant offset (red arrow). The diagram is calculated with a microswimmer efficiency of $\eta_m = 0.4$.

autocorrelation function

$$\langle \boldsymbol{\xi}(t) \boldsymbol{\xi}(t + \tau) \rangle = 2\mathbf{D}\delta(\tau) \tag{40}$$

where the diffusion tensor follows from the Stokes–Einstein relationship as $\mathbf{D} = k_B T \mathbf{R}_V^{-1}$. Here we disregard any additional active sources of noise resulting from the stochasticity of the propulsion mechanism.

The entropy production of the above particle in a steady state is commonly referred to as "housekeeping" entropy[50]. In the classical picture that replaces the self-propulsion velocity $\mathbf{V}_A^0$ with the motion due to an active force, $\mathbf{R}_V^{-1} \cdot \mathbf{F}_A^0$, the mean entropy production can be expressed from the statistical definition as in refs. [16–18]. For translational motion, it can be written as $T\dot{S}_{\text{hk}} = \langle \dot{\mathbf{X}} \cdot (\mathbf{F} + \mathbf{R}_V \cdot \mathbf{V}_A^0) \rangle$ where $\dot{\mathbf{X}}$ is the stochastic velocity (38) and $\mathbf{R}_V \cdot \mathbf{V}_A = \mathbf{F} + \mathbf{R}_V \cdot \mathbf{V}_A^0$ the deterministic total force, consisting of the external and the active contribution. The brackets indicate averaging over noise realizations, as well as over particle positions and orientations. By carrying out the noise average while keeping the position and orientation constant, $\langle \dot{\mathbf{X}} \rangle_{\mathbf{X}, \mathbf{n}} = \mathbf{R}_V^{-1} \cdot \mathbf{F} + \mathbf{V}_A^0$, the translational contribution to $T\dot{S}_{\text{hk}}$ becomes $\langle \mathbf{V}_A \cdot \mathbf{R}_V \cdot \mathbf{V}_A \rangle$. A similar result can be obtained for rotational motion, as shown in ref. [18] for a single degree of freedom. Taken together, the total "housekeeping" entropy production in our six-component notation reads

$$T\dot{S}_{\text{hk}} = \langle \mathbf{V}_A \cdot \mathbf{R}_V \cdot \mathbf{V}_A \rangle \tag{41}$$

with the brackets indicating averaging over particle positions and orientations.

At this point, we apply the finding (36) that the minimum dissipation rate by the active swimmer can be written as a sum of a constant and a term proportional to the square of the actual velocity (swimming velocity plus drift caused by external forces). The second term is identical to the expression in Eq. (41). By adding the zero-velocity dissipation (37), the total "housekeeping" entropy production increases to

$$T\dot{S}_{\text{hk}}^{\text{tot}} \geq \mathbf{V}_A^0 \cdot \left[ \left( \mathbf{R}_V^{-1} - \mathbf{R}_F^{-1} \right)^{-1} - \mathbf{R}_V \right] \cdot \mathbf{V}_A^0 + \langle \mathbf{V}_A \cdot \mathbf{R}_V \cdot \mathbf{V}_A \rangle \tag{42}$$

The expression holds for any combination of translational and rotational motion. For free-swimming particles, the housekeeping entropy production can be expressed by means of the microswimmer efficiency, as defined in ref. [38]

$$\eta_m = \frac{\mathbf{V}_A^0 \cdot \mathbf{R}_V \cdot \mathbf{V}_A^0}{\mathbf{V}_A^0 \cdot \left( \mathbf{R}_V^{-1} - \mathbf{R}_F^{-1} \right)^{-1} \cdot \mathbf{V}_A^0}. \tag{43}$$

Its lower bound is then given by

$$T\dot{S}_{\text{hk}}^{\text{tot}} \geq \frac{1}{\eta_m} \mathbf{V}_A^0 \cdot \mathbf{R}_V \cdot \mathbf{V}_A^0. \tag{44}$$

Our expression points to a fundamental limit on dissipation by active particles that is stricter than obtained by treating them as if they were passive particles with an external driving force $\mathbf{R}_V \cdot \mathbf{V}_A$ and a drag coefficient $\mathbf{R}_V$, because the housekeeping dissipation needs to be offset by the contribution of Eq. (37).

## Discussion

In this study, we derived fundamental limits on dissipation by several classes of microswimmers with internal dissipation. The scenarios we discussed describe the major propulsion mechanisms by different microswimmers: active droplets driven by the Marangoni effect[43], ciliated microswimmers that can be approximated with an extended force-generating layer[44], phoretic swimmers with a position-dependent zeta potential, and finally a coarse-grained model with an arbitrary local dissipation density.

Determining a lower bound on the dissipation requires finding the distribution of forces that minimize dissipation while maintaining a given swimming speed for a given type and shape of swimmer. In principle, this presents a complex PDE-constrained optimization problem[36]. However, it can be solved by generalizing a very powerful approach previously derived for swimmers with external dissipation alone, i.e., swimmers with an idealized, lossless, propulsion mechanism[38]. The original minimum dissipation theorem allowed us to express the minimum external dissipation by a microswimmer with two passive drag coefficients—one of a no-slip and one with a perfect-slip body of the same shape. The derivation of the theorem was based on two properties of the Stokes flow: the Helmholtz minimum dissipation theorem[51] and the Lorentz reciprocal theorem[40]. Here, we show that under certain conditions analogous theorems can be derived for other swimmer models that do take into account internal dissipation. Specifically, one needs to find two passive problems that satisfy the velocity boundary conditions of the original problem. One of them needs to represent the flow with minimum dissipation under these boundary conditions and the other one is a passive problem with velocities orthogonal to the active forces. The lower bound on dissipation can then be expressed with the reciprocal difference between the two drag coefficients. Although we restricted our discussion to five scenarios, we expect that other minimum dissipation problems can be solved with the same method. A straightforward example is a microswimmer with a propulsive layer covering only a part of its surface. We also expect that the theorem can be generalized to other propulsion mechanisms, for example, surface tension around a swimmer embedded in a liquid-air interface ("Marangoni surfer"[52,53]). Note that besides determining the bound on dissipation, the theorem, by means of linear superposition, also provides a distribution of forces that exactly achieves this bound, thus fully solving the optimization problem.

We subsequently expanded the optimality to a class of swimmers with a fixed volume, but different shapes. The optimal shape of a surface-driven droplet depends on the ratio between the internal–external viscosity. When the internal viscosity is small, the optimal shape unsurprisingly becomes prolate and eventually needle-like, in order to minimize the external dissipation. On the other hand, if the internal viscosity is large, the optimal shape undergoes a topological bifurcation and takes the shape of a toroid rotating inside-out, resembling some swimmer models studied by Taylor[47] and Purcell[11]. Swimmers with effective surface dissipation always become elongated and, like in the case of surface dissipation alone, take the shape of a body with two protrusions along the symmetry axis if sufficient

curvature is permitted. These shapes bear a remarkable resemblance with many ciliates found in nature[24].

When the swimmer is moving in the presence of an external force, the total dissipation can be decomposed into a constant term and a term that corresponds to the drag of a passive body moved with the same velocity. The dissipation therefore reaches its minimum when the swimmer is stalled. The drag coefficient that appears in the expression for dissipation is the same drag coefficient that describes the velocity response to an external force and also determines the noise amplitude through the fluctuation-response theorem. In contrast with the commonly used picture in which the propulsion of a microswimmer is replaced by an external "active force"[14,16,17], we find that the dissipation and entropy production need to be corrected for a contribution describing the energetic cost of force generation. The same holds for studies that determine the entropy production from the statistical definition[18,54,55]. Our work therefore complements the previous studies on fundamental limits on entropy production in active microswimmer suspensions by adding an unavoidable contribution from internal dissipation and also from the fact that swimming usually generates more dissipation than pulling by a force. It needs to be stressed that these results hold under the assumption that the presence of an external load does not affect the active driving forces on the swimmer. In principle, other types of response are also conceivable. For example, the opposite limit would be a mechanism maintaining a prescribed slip velocity regardless of the stress on the surface. In such cases, the velocity-dependent entropy production rate can have a different form. Furthermore, the entropy production can be influenced by interactions between swimmers, in particular when they are non-conservative[56].

Finally, while our study can provide a complete hydrodynamic picture of the problem, it does not take into account the dissipation by the mechanism of force production, for example, a chemical reaction in a phoretic swimmer. The energetics of the latter were addressed in several studies[19,29], but a complete solution providing a dissipation limit for the combined chemical and hydrodynamic problem still presents an open challenge. Likewise, a major open question is whether our approach can be used to provide dissipation limits for swimmers that move by periodically changing their shape, which was thus far possible only under limited constraints (e.g., the three-sphere swimmer[57]).

## Methods

### Lorentz reciprocal theorem

In the following, we recapitulate the Lorentz reciprocal theorem[40,58], which provides us a relationship between two different flow problems sharing the same geometry and fluid medium. The main problem (here denoted as A) and the auxiliary problem (B) are then connected through the following integral relationship:

$$\int_{\mathcal{S}} dS\, \mathbf{f}_A \cdot \mathbf{v}_B + \int_{\mathcal{V}} dV\, \mathfrak{f}_A \cdot \mathbf{v}_B = \int_{\mathcal{S}} dS\, \mathbf{f}_B \cdot \mathbf{v}_A + \int_{\mathcal{V}} dV\, \mathfrak{f}_B \cdot \mathbf{v}_A. \quad (45)$$

Here $\mathbf{f}$ denotes the traction on the integration surface and $\mathfrak{f}$ any additional body force in the integration volume.

If problem A consists of a body moving with the rigid body velocity $\mathbf{V}_A$ and problem B of a body with the same shape moving with $\mathbf{V}_B$, the reciprocal theorem can also be expressed with co-moving velocities

$$\int_{\mathcal{S}} dS\, \mathbf{f}_A \cdot \bar{\mathbf{v}}_B + \int_{\mathcal{V}} dV\, \mathfrak{f}_A \cdot \bar{\mathbf{v}}_B + \mathbf{F}_A \cdot \mathbf{V}_B = \int_{\mathcal{S}} dS\, \mathbf{f}_B \cdot \bar{\mathbf{v}}_A + \int_{\mathcal{V}} dV\, \mathfrak{f}_B \cdot \bar{\mathbf{v}}_A + \mathbf{F}_B \cdot \mathbf{V}_A. \quad (46)$$

A classical application is to apply the reciprocal theorem in this form to an active force-free swimmer ($\mathbf{F}_A = 0$) and a no-slip body (B ≡ NS) of the

same shape, which satisfies $\bar{\mathbf{v}}_B = \bar{\mathbf{v}}_{NS} = 0$. Then the left-hand side of Eq. (46) is zero and the right-hand side yields $\mathbf{F}_{NS} \cdot \mathbf{V}_A = -\int_{\mathcal{S}} dS\, \mathbf{f}_{NS} \cdot \bar{\mathbf{v}}_A$, which is an elegant and frequently used way of determining the velocity of the active swimmer $\mathbf{V}_A$ if one knows its surface velocity $\bar{\mathbf{v}}_A$[30,40,59].

## Data availability

All data required to generate our results are available from the corresponding author upon request.

## Code availability

The code used to obtain the optimal shapes of active droplets and swimmers with surface dissipation (Figs. 4 and 5) is available from the corresponding author upon request.

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

## Acknowledgements

We thank Benoît Mahault for insightful comments on the manuscript and Ray Goldstein for drawing our attention to the core–shell model of *Volvox* which inspired the tangentially driven swimmer model. We acknowledge support from the Max Planck Center Twente for Complex Fluid Dynamics (A.D.-M.-I.), the Max Planck School Matter to Life, and the MaxSynBio Consortium, which are jointly funded by the Federal Ministry of Education and Research (BMBF) of Germany (R.G.), the Max Planck Society (A.D.-M.-I., R.G., and A.V.) and the Slovenian Research Agency (grant no. P1-0099, A.V.).

## Author contributions

R.G. and A.V. designed research; A.D.-M.-I and A.V. performed research; and A.D.-M.-I., R.G., and A.V. wrote the paper.

## Funding

## Competing interests

The authors declare no competing interests.
