## [Peer Review File · Nature Communications]

Minimum Entropy Production by Microswimmers with Internal DissipationREVIEWER COMMENTS

Reviewer #1 (Remarks to the Author):

Vilfan and colleagues present a general theory regarding the minimal entropy production of microswimmers. Through a clever manipulation and extension of two theorems of low Reynolds number hydrodynamics, they are able to derive general bounds for several classes of microswimmers nicely summarized in Table 1 and Figure 2.

The derivations and results substantially extend what they had presented earlier in their 2021 Phys. Rev. Lett. (Ref. 37) even though the basic strategy is the same: One needs to solve two passive problems with specific boundary conditions whose combination yields the bound on the active one.

As main new features not yet treated in Ref. 37, they allow for dissipation in a boundary layer, or, more general, internal dissipation and motion under additional external forces.

The paper is very well written and the results are presented in a systematic way. I would expect this paper to become a reference for quite some time for anyone studying dissipation for microswimmers. In my opinion, this justifies publication in a high profile journal like Nat. Comm..

In a minor revision, the following issues should be addressed.

1. Figure 6: Since none of the axes carries a scale, it is not clear where the stated value of $\eta_m = 0.4$ enters. How would it look like for, say, $\eta_m = 0.8$?
2. The discussion around eqs. 40-42 is too brief and too vague. In which sense does the expression in eq. 42 "point to the inadequacy" of the earlier approach?

Reviewer #2 (Remarks to the Author):

The manuscript reports on a theoretical examination of microswimmers' motion in the limit of zero Reynolds number, where no inertial effects are present. In particular, a theoretical framework is developed to study the energy expenditure and entropy generation during the self-propulsion of active particles. The theory, which builds on the authors' previous works on the same topic, is applied to bulk swimmers with several different propulsion mechanisms to calculate, given the same volume, the shapes resulting in minimum energy dissipation. Overall, the presented work is a valuable theoretical addition to the literature on the hydrodynamics of microswimmers. My comments for improving the quality of the manuscript are:

- The manuscript can benefit from a subsection where the mathematical problem is clearly stated and the relevant variables are defined.
- It would be interesting to discuss how the derived theorem can be extended to include active particles that move along liquid-air interfaces, e.g., Marangoni surfers.

Reviewer #3 (Remarks to the Author):

The authors study the energy dissipation of self-propelled microswimmers. Their main results are lower bounds for the energy dissipation of such active swimmers, obtained by an ingenious combination of two flow configurations around related passive (i.e. externally driven) particles. These bounds are derived for a various different types of swimmers (Table I), and are used to derive optimal swimmer shapes that minimize dissipation. I think this is an excellent and very important work providing bounds for the entropy production of swimming at low Reynolds numbers on quite general grounds. Also, the manuscript is written very well and should be accessible to a broad readership.

Nevertheless, I have a few comments that might help to further improve the manuscript, and also some questions on points that are not completely clear to me after studying the manuscript:

- 1) The superscript $\|\|$ in the text above eq (2) and in eq (4) is not defined. (It is clear what it means, but probably should be stated anyway for the sake of completeness.)
- 2) I think it would be useful to include (maybe in Section B of the SI) more background on the resistance tensors, as these are the central quantities entering the dissipation bounds. For instance, short descriptions of how the resistance tensor is defined, how it is related to the dissipation rate (S.6) how it can be calculated (in principle).
- 3) It is not obvious to me that the difference of the resistance tensors appearing in the dissipation bounds is still positive definite, such that the bounds are actually non-trivial (since dissipation is always positive). Can this be demonstrated?
- 4) I am not sure if I fully understand what kind of dissipation is included in the "internal dissipation". As far as I understand, the internal dissipation quantifies the dissipation due to hydrodynamic stresses in the propulsive layer, which result from the propulsion mechanism (which is not further specified) creating a specific local "propulsion flow field"; the latter is characterized by a specific set of boundary conditions. However, it does not include any dissipation connected to the creation of the flow field itself. Is that correct? Maybe the authors can elaborate a little bit on more on these points. On that note: why does the specific prefactor in the second term of eq (3) characterize internal dissipation? What is μ here?
- 5) About boundary conditions: I suggest that the authors include a table listing all the different boundary conditions (as equations) used throughout the manuscript, and to connect them to the terminology used in Table I.
- 6) Below eq (44), I suggest to state explicitly that the no-slip boundary conditions are identified with the Stokes-problem in (44).
- 7) Concerning the swimmers with the lowest dissipation: As the authors explain, it is known from their calculations how the flow field around the swimmer has to look like to minimize dissipation. Thus, to find the swimmer with minimal dissipation, does one has to change its shape until it "produces" (together with the boundary conditions) this specific flow field? Also, when changing the shapes (like in Fig. 5), does one have to adjust the size of the outer surface?
- 8) I do not understand the discussion in Sec. F of the manuscript. Specifically, are not all quantities entering in eq (40) deterministic? I suspect that V_A is an average velocity? What

does then the average mean? Moreover, is not the housekeeping dissipation discussed here connected to the external dissipation? I think in the case of passive particles they are equivalent. For active swimmers, there might be parts of (40) already contained in (37). Finally, how to define "entropy" production and dissipation from the equations of motion (38) is not immediately obvious for active particles, because of their inherent non-equilibrium nature. In fact, simply using the standard expressions of stochastic thermodynamics does not give the entropy produced in the thermal bath (or the heat dissipated into the thermal bath); the authors cite the pertinent literature.

In summary I think this manuscript represents an essential contribution to understanding dissipation for active microswimmers, and I recommend its publication in Nature Communications. Before publication, the authors might want to consider the points listed above.

Reviewer #4 (Remarks to the Author):

This an interesting paper dealing with minimum dissipation bounds for a variety of swimmers. The authors build on their previous results in Ref 37 to construct minimum dissipation bounds in a variety of regimes. Towards the end of the manuscript, they also consider the stochastic thermodynamic analogue of this problem including terms like housekeeping heat. The paper is well written and seems to be technically sound. My main concern at this point is (a) I am not sure if there is enough novelty beyond what was considered in Ref 37 to merit publication in a journal like Nat Comm. (B) I don't completely see the point of the stochastic thermodynamic analysis and how it contributes. The authors mention the inadequacy of an effective approach in which the active particles in a medium are treated as if they have a constant force applied on them. But this point is well appreciated already.

In summary, I think this is a very nice piece of technical work that deserves publication in a specialized journal.

Response to Reviewer #1:
(with Reviewer's comments in blue)

Vilfan and colleagues present a general theory regarding the minimal entropy production of microswimmers. Through a clever manipulation and extension of two theorems of low Reynolds number hydrodynamics, they are able to derive general bounds for several classes of microswimmers nicely summarized in Table 1 and Figure 2.

The derivations and results substantially extend what they had presented earlier in their 2021 Phys. Rev. Lett. (Ref. 37) even though the basic strategy is the same: One needs to solve two passive problems with specific boundary conditions whose combination yields the bound on the active one.

As main new features not yet treated in Ref. 37, they allow for dissipation in a boundary layer, or, more general, internal dissipation and motion under additional external forces.

The paper is very well written and the results are presented in a systematic way. I would expect this paper to become a reference for quite some time for anyone studying dissipation for microswimmers. In my opinion, this justifies publication in a high profile journal like Nat. Comm..

We thank the Reviewer for the careful evaluation of our manuscript and for identifying points that need clarification in the last section. In the following, we address the two specific comments made by the Reviewer:

In a minor revision, the following issues should be addressed.

1. Figure 6: Since none of the axes carries a scale, it is not clear where the stated value of $\eta_m=0.4$ enters. How would it look like for, say, $\eta_m=0.8$?

Here, η_m does, as the only parameter, affect the scale-free shape of the diagram. According to Eq. 42, $1/\eta_m$ is the ratio between the dissipation of a force-free swimmer (red dot) and the expression $R_v V_0^2$ (gray line underneath the red dot, i.e., $V=V_0$). For $\eta_m=0.8$ they would be relatively closer together, see below:

We have now changed the misleading wording “the values show”, because the graph indeed shows no values. We replaced it with “The diagram is calculated with ...”.

2. The discussion around eqs. 40-42 is too brief and too vague. In which sense does the expression in eq. 42 "point to the inadequacy" of the earlier approach?

This comment overlaps with comment (8) by Reviewer #3.

We see that this derivation deserves a better explanation and we have largely extended it with the revision. By inadequacy we mean that the classical papers in the field only consider the term $R_v V^2$, whereas we show that the actual dissipation limit is higher by a factor $1/\eta_m$. We hope that the revised text makes the derivation clearer.

Response to Reviewer #2:
(with Reviewer's comments in blue)

The manuscript reports on a theoretical examination of microswimmers' motion in the limit of zero Reynolds number, where no inertial effects are present. In particular, a theoretical framework is developed to study the energy expenditure and entropy generation during the self-propulsion of active particles. The theory, which builds on the authors' previous works on the same topic, is applied to bulk swimmers with several different propulsion mechanisms to calculate, given the same volume, the shapes resulting in minimum energy dissipation. Overall, the presented work is a valuable theoretical addition to the literature on the hydrodynamics of microswimmers.

We thank the Reviewer for the careful reading of our manuscript and for the insightful comments, dealing with a gap in the presentation and also bringing up a highly interesting proposition. We have revised our manuscript according to the Reviewer's comments as detailed below.

My comments for improving the quality of the manuscript are:

- The manuscript can benefit from a subsection where the mathematical problem is clearly stated and the relevant variables are defined.

We agree that we took too many aspects of low Reynolds number swimmers for granted in the original manuscript. We have therefore included a passage introducing the Stokes equation and the microswimmer at the beginning of Section A of the Results. We also moved there some definitions from other parts of the manuscript, like velocities in the lab- and co-moving frame.

- It would be interesting to discuss how the derived theorem can be extended to include active particles that move along liquid-air interfaces, e.g., Marangoni surfers.

In principle the interface can easily be included, because the generalized Helmholtz minimum dissipation theorem also holds for the flow around a particle immersed in the interface. Marangoni surfers are particles that achieve propulsion by modulating the surface tension in the interface around them. The complication here is that the forces acting on the fluid (and oppositely on the particle) cannot be adjusted freely, but need to be gradients of a scalar function (surface tension). To derive a minimum dissipation theorem, we therefore need a passive problem with curl-free tractions on the surface and a velocity distribution that is "orthogonal" to the traction in the Marangoni flow. Our preliminary calculations show that this is the case if the "F-problem" is a particle embedded in a fluid with an agent that ensures that the surface velocity is divergence free. In other words, the surface area of the fluid needs to be locally conserved. We are excited about the idea and very grateful to the Reviewer for bringing it up. We now mention this idea in the discussion among future applications and cite one theoretical and one recent experimental reference, but we think that the solution will require (and deserve) a separate paper.

Response to Reviewer #3:
(with Reviewer's comments in blue)

The authors study the energy dissipation of self-propelled microswimmers. Their main results are lower bounds for the energy dissipation of such active swimmers, obtained by an ingenious combination of two flow configurations around related passive (i.e. externally driven) particles. These bounds are derived for a various different types of swimmers (Table I), and are used to derive optimal swimmer shapes that minimize dissipation. I think this is an excellent and very important work providing bounds for the entropy production of swimming at low Reynolds numbers on quite general grounds. Also, the manuscript is written very well and should be accessible to a broad readership.

We thank the Reviewer for the attentive reading of our manuscript and for the carefully assembled list of points in the presentation that require improvement or clarification. We have revised our manuscript according to the Reviewer's comments as detailed below.

Nevertheless, I have a few comments that might help to further improve the manuscript, and also some questions on points that are not completely clear to me after studying the manuscript:

1) The superscript \parallel in the text above eq (2) and in eq (4) is not defined. (It is clear what it means, but probably should be stated anyway for the sake of completeness.)

We agree that it should be defined at first appearance. We have therefore included a definition in the paragraph above Eq. (1).

2) I think it would be useful to include (maybe in Section B of the SI) more background on the resistance tensors, as these are the central quantities entering the dissipation bounds. For instance, short descriptions of how the resistance tensor is defined, how it is related to the dissipation rate (S.6) how it can be calculated (in principle).

Following the Reviewer's suggestion, we added a paragraph after Eq. S8 in which we define the generalized drag coefficients, show their composition of translational/rotational components and show how they are related to the dissipation rate.

3) It is not obvious to me that the difference of the resistance tensors appearing in the dissipation bounds is still positive definite, such that the bounds are actually non-trivial (since dissipation is always positive). Can this be demonstrated?

Yes! We thank the Reviewer for bringing up this interesting question. In each of the cases, the passive minimum dissipation theorem for the V-problem has fewer constraints than the F-problem. For example, the modified Helmholtz theorem states that among all bodies with zero normal velocity on the surface (impermeability), the perfect-slip body ("bubble") has the smallest possible drag. The F-problem (no-slip boundary) has an extra constraint, namely that the tangential velocity also be zero. It will always have a higher drag coefficient (higher dissipation at the same velocity). The same holds for all other types of bodies discussed in the manuscript. Knowing that the difference $\mathbf{R}_F - \mathbf{R}_V$ is positive definite and so are \mathbf{R}_F and \mathbf{R}_V themselves (by the nature of a drag coefficient), one can show that the RHS of our theorem (Eq. 1) is also positive definite. We have now included this explanation in the paragraph following Eq. (2) and commented on it for other inequalities.

We also note that the bounds derived in this paper are all achievable in principle (as far as hydrodynamic losses are concerned), so they must be positive definite.

4) I am not sure if I fully understand what kind of dissipation is included in the "internal dissipation". As far as I understand, the internal dissipation quantifies the dissipation due to hydrodynamic stresses in the propulsive layer, which result from the propulsion mechanism (which is not further specified) creating a specific local "propulsion flow field"; the latter is characterized by a specific set of boundary conditions. However, it does not include any dissipation connected to the creation of the flow field itself. Is that correct?

Maybe the authors can elaborate a little bit on more on these points. On that note: why does the specific prefactor in the second term of eq (3) characterize internal dissipation? What is μ here?

In general, the internal dissipation can be any source of dissipation that is proportional to the square of the slip velocity. The theorem that follows Eq. (3) holds for any flow, active or passive, with or without external force on the swimmer. μ in Eq. (3) is the viscosity of the surrounding fluid; in principle μ/λ could be any constant here (or even a scalar function of position on the surface). We did, however, write it as μ/λ in anticipation of the result that dissipation is minimized with the Navier slip condition, where λ becomes the slip length. We have included these explanations in the text that follows Eq. (3).

5) About boundary conditions: I suggest that the authors include a table listing all the different boundary conditions (as equations) used throughout the manuscript, and to connect them to the terminology used in Table I.

We have now included Table II with the boundary conditions.

6) Below eq (44), I suggest to state explicitly that the no-slip boundary conditions are identified with the B-problem in (44).

We have added the statement “B \equiv NS”, as well as $\mathbf{v}_B = \mathbf{v}_{NS} = 0$, in the text below Eq. (44).

7) Concerning the swimmers with the lowest dissipation: As the authors explain, it is known from their calculations how the flow field around the swimmer has to look like to minimize dissipation. Thus, to find the swimmer with minimal dissipation, does one has to change its shape until it "produces" (together with the boundary conditions) this specific flow field? Also, when changing the shapes (like in Fig. 5), does one have to adjust the size of the outer surface?

Our theorem provides the optimal flow field only for a swimmer of a fixed shape. In the shape optimization procedure, an explicit flow calculation is not needed, because the optimal efficiency can be expressed with the two drag coefficients. We expanded the corresponding explanation.

The optimization is carried out under the condition of a fixed volume, therefore as the shape is changed, the body is immediately rescaled to the right volume. This automatically adjusts the size of the surface. We have now stated explicitly that each shape is rescaled to unit volume before calculating the dissipation.

8) I do not understand the discussion in Sec. F of the manuscript. Specifically, are not all quantities entering in eq (40) deterministic? I suspect that \mathbf{V}_A is an average velocity? What does then the average mean?

The issues addressed are also related to comment 2 by Reviewer #1.

We have now expanded the discussion to explain the meaning of different averages. For a given position and orientation of the particle, \mathbf{V}_A is deterministic (i.e., it is the velocity averaged over all noise realizations). The average represents the weighted average of all particle positions and orientations (or time average of a single particle). For example, in a harmonic potential trap, particles swimming away from the trap center will have a smaller dissipation than those swimming towards the center.

Moreover, is not the housekeeping dissipation discussed here connected to the external dissipation? I think in the case of passive particles they are equivalent.

Equation (40) refers to the “traditional” picture where active swimming is replaced by effective external forces, so the housekeeping entropy production does correspond to the work of those forces, as the Reviewer writes. In the expanded derivation, we point this out more clearly.

For active swimmers, there might be parts of (40) already contained in (37).

Equation (36) splits the dissipation into a velocity-dependent contribution (that is identical to the one that appears in Eq. 40) and a constant part. The constant part is later evaluated in Eq. 37. From this

decomposition, we conclude that the part in Eq. (37) appears as an addition when the dissipation of the active swimmer is considered.

Finally, how to define "entropy" production and dissipation from the equations of motion (38) is not immediately obvious for active particles, because of their inherent non-equilibrium nature. In fact, simply using the standard expressions of stochastic thermodynamics does not give the entropy produced in the thermal bath (or the heat dissipated into the thermal bath); the authors cite the pertinent literature.

We naturally agree that evaluating the entropy production from the equations of motion is not straightforward, which is why we take the detour via driven particles. We also note that our results are limited to the housekeeping (steady state mean) entropy production and that we can only provide a lower bound from the hydrodynamic perspective, but under these conditions it should correspond to the entropy increase of the thermal bath.

In summary I think this manuscript represents an essential contribution to understanding dissipation for active microswimmers, and I recommend its publication in Nature Communications. Before publication, the authors might want to consider the points listed above.

Response to Reviewer #4:

(with Reviewer's comments in blue)

This an interesting paper dealing with minimum dissipation bounds for a variety of swimmers. The authors build on their previous results in Ref 37 to construct minimum dissipation bounds in a variety of regimes. Towards the end of the manuscript, they also consider the stochastic thermodynamic analogue of this problem including terms like housekeeping heat. The paper is well written and seems to be technically sound. My main concern at this point is (a) I am not sure if there is enough novelty beyond what was considered in Ref 37 to merit publication in a journal like Nat Comm. (B) I don't completely see the point of the stochastic thermodynamic analysis and how it contributes. The authors mention the inadequacy of an effective approach in which the active particles in a medium are treated as if they have a constant force applied on them. But this point is well appreciated already.

In summary, I think this is a very nice piece of technical work that deserves publication in a specialized journal.

We thank the Reviewer for the critical assessment of our manuscript. The Reviewer addresses two important questions about our manuscript:

- (a) Novelty with respect to our 2021 PRL where we derived a minimum dissipation theorem for external dissipation alone and
- (b) The relevance of the findings for the entropy production by active particles.

We would like to provide the following response regarding the two issues raised by the Reviewer:

(a) We are convinced that the work only appears as an extension of Ref [37] in hindsight. We naturally invested a lot of effort into a clear derivation that looks "straightforward" – now. Two years ago, we considered the solution in that paper as a one-off and the way towards including internal dissipation was far from being anticipated, neither by us nor by anybody we discussed the results with.

At the same time, including internal dissipation means a leap from a theoretical lower bound that is only achievable in a very hypothetical scenario, to an inequality that takes into account the reality of different physical microswimmer concepts. We therefore think that the applicability of our theory is now at a completely different level.

(b) Although the difference between active swimmers and externally pulled particles is in principle well known, the body of literature on entropy production by microswimmers treats them using external forces (see references in the paper). Providing a physically consistent lower bound on the dissipation is in our opinion a way of giving this field a solid foundation.

REVIEWER COMMENTS

Reviewer #1 (Remarks to the Author):

I am fully satisfied with the revision and the answer to the requests of the referees. I am happy to recommend publication of this nice manuscript as now is.

Reviewer #2 (Remarks to the Author):

The authors' responses are satisfactory.

Reviewer #3 (Remarks to the Author):

The authors took into account all the comments of the various referees, and revised their manuscript accordingly. I find the revisions satisfactory, except concerning the discussions in Sec. F. There are still a few points which are unclear to me:

- * It is not clear to me, why the authors do not directly calculate the power from the full flow field, including the power contributed by the external force, i.e. including it directly in (S.17)
- * Then one could compare with the dissipation rate obtained in the "stochastic thermodynamics way" based on the equation of motion (38) in which the self-propulsion is mimicked by an additional force, and demonstrate that the latter does not give the correct entropy production.
- * In the paragraph between eq (39) and eq (49), there is a quantity V_0 ; I suppose it should be V_A ?
- * The averaging procedure is presented in a slightly imprecise way: the orientational degrees of freedom are also affected by the thermal noise, i.e. when carrying out the noise average $\langle \dot{X} \rangle$ there remains an average on the right-hand side over the rotational noise components. Like written now (just before eq (40)), it is an average only over the translational noise, i.e. conditioned on fixed orientation. Likewise, maybe it is useful to specify that V_A is $\langle \dot{X} \rangle_{\text{fixed orientation}}$, i.e. a noise average at fixed orientation. As a consequence, the average in (40) is over the orientational noise.
- * What about the contributions of the rotational motion to the dissipation and entropy production rate? It might be useful to briefly comment on that.

I suggest a minor revision of the manuscript along these lines before publication in Nature Communications, which I recommend enthusiastically.

Response to Reviewer #3

We thank the Reviewer for the careful reading of our revised manuscript and for constructive comments about the derivation in Section F. We have taken them all into account in the revised version.

In the following, we are giving a point-by-point response to the Reviewer's comments (quoted in blue). Please note that equation numbers from (35) onwards are shifted down by one in the revision:

The authors took into account all the comments of the various referees, and revised their manuscript accordingly. I find the revisions satisfactory, except concerning the discussions in Sec. F. There are still a few points which are unclear to me:

* It is not clear to me, why the authors do not directly calculate the power from the full flow field, including the power contributed by the external force, i.e. including it directly in (S.17)

We appreciate this comment – we agree that the number of different measures for the power/dissipation can be confusing and that we can skip the introduction of the power contributed by the propulsion mechanism P_A . We have therefore shortened the derivation and also changed the nomenclature such that P_A now denotes the total dissipated power throughout the paper. The Supplementary information is also adjusted accordingly. We only show the split into the propulsive power and the rate of work by the external force in Fig. 6.

* Then one could compare with the dissipation rate obtained in the "stochastic thermodynamics way" based on the equation of motion (38) in which the self-propulsion is mimicked by an additional force, and demonstrate that the latter does not give the correct entropy production.

In the revised manuscript, this becomes evident when comparing the equations (new numbering) (35) and (39).

* In the paragraph between eq (39) and eq (49), there is a quantity V_0 ; I suppose it should be V_A^0 ?

We agree that the notation should be unified, also when writing about models from the literature. We have therefore replaced V_0 with V_A^0 as suggested.

* The averaging procedure is presented in a slightly imprecise way: the orientational degrees of freedom are also affected by the thermal noise, i.e. when carrying out the noise average $\langle \dot{X} \rangle$ there remains an average on the right-hand side over the rotational noise components. Like written now (just before eq (40)), it is an average only over the translational noise, i.e. conditioned on fixed orientation. Likewise, maybe it is useful to specify that V_A is $\langle \dot{X} \rangle_{\text{fixed orientation}}$, i.e. a noise average at fixed orientation. As a consequence, the average in (40) is over the orientational noise.

We agree that the fixed orientation (and position) should be denoted. In the revised manuscript we are using the notation as in Ref. [18] (Eqs. 10 and 11 therein) for the average.

* What about the contributions of the rotational motion to the dissipation and entropy production rate? It might be useful to briefly comment on that.

We note that although the part dealing with substitute external active forces writes about translational motion only, all our derivations work with 6-dimensional generalized velocities and

therefore equally include rotational contributions. We now comment on that in the derivation, with a reference to [18]. In addition, we comment on its validity for any combination of translational and rotational motion after Eq. (40).

REVIEWERS' COMMENTS

Reviewer #3 (Remarks to the Author):

The authors have clarified the points I raised in my comments, the corresponding revisions in the manuscript and the SI are fine.

I am happy to see the manuscript published in Nature Communications soon.